# Laboratory Rat Thrombi Lose One-Third of Their Stiffness When Exposed to Large Oscillating Shear Stress Amplitudes: Contrasting Behavior to Human Clots

**Ursula Windberger** [1],*, **Veronika Glanz** [2] **and Leon Ploszczanski** [3]

1    Decentralized Biomedical Facilities, Core Unit Laboratory Animal Breeding and Husbandry,
     Medical University Vienna, 1090 Vienna, Austria
2    General Intensive Care and Pain Management, Department of Anaesthesia, Medical University Vienna,
     1090 Vienna, Austria; veronika.glanz@meduniwien.ac.at
3    Institute of Physics and Materials Science, University of Natural Resources and Life Sciences,
     1190 Vienna, Austria; leon.ploszczanski@boku.ac.at
*    Correspondence: ursula.windberger@meduniwien.ac.at

**Abstract:** Rats impress by their high platelet count resulting in hypercoagulability, which protects the animals from severe bleeding. However, platelets also import numerous stiff junction points into the fibrous system of a clot, also enhancing the pre-stress of the fibrin fibers, which lowers their deformability. Clot deformation is clinically important since large strains are present in the arterial tree (caused by the propagation of pressure and pulse waves), and a clot is considered "safe" when it can deform over a long range of strain amplitudes. We tested clot formation and the behavior of fully formed blood clots of laboratory rats at large sinusoidal shear stress amplitudes by rheometry and compared outcomes to human reference data. We found that fiber density (by scanning electron microscopy) and clot stiffness (by rheometry) was pronounced compared to humans and differed with sexual dimorphism and with rat strain. Using our large amplitude oscillation (LAOS) protocol, we detected that rat clots yielded with a frustrated attempt to stiffen instead of showing the macroscopic stiffening response that is typical for human clots. We attribute this behavior to the appearance of multiple microfractures until, finally, a few leading fibers uptake the load. Rat clots also failed to align fibers in shear direction to initiate affine deformation. The rat clot phenotype differs substantially from the human one, which must be considered in research and toxicological testing. If microfractures in the fiber meshwork are concentrated in vivo, parts of a clot may break off and be washed away. However, homogenously distributed microfractures may open pores and allow the penetration of plasminogen activators. What occurs in the rat vasculature depends on the on-site clot composition.

**Keywords:** rat blood clot; yielding; LAOS; rat strains; clot stiffness; clot deformation; rheometry; thrombelastometry

## 1. Introduction

Rat strains are widely used in coagulation studies, although there are significant differences in blood coagulation compared to the human standard [1]. Although the general principle of blood coagulation, clot formation, clot remodeling, and fibrinolysis are provided among species, there exist species-specific differences, mostly based on the structure or quantity of the proteins involved (for review see [2]). The most striking difference between human and rat blood coagulation lies in the short clotting times in rats to values that would be pathological for human blood [3–6]. An assembly of factors contributes to this. Among them are a higher concentration of circulating clotting factors (V, II, XII, XIII), but some factors (X, VIIIR:vW, IX) were assayed lower in rat than in human plasma [1,3,7,8]. Rat platelets contain more myosin but less actin than human platelets [9], which will affect clot contraction. Some rat strains display platelet hyperreactivity [10], and

the plasma fibrinolytic system has a higher resistance to activation [11]. On other hand, in the murine system thrombin binds less to the GpIb platelet receptor [5,12], platelets are irresponsive to the thrombin receptor activating peptide [13], and platelet aggregation cannot be triggered by adrenalin [14]. Thrombin and plasmin generation differed from humans and were reduced [15]. Plasminogen was shown to be activated by human urokinase but required the presence of a human proactivator for streptokinase activation [8]. Due to such differences, it was postulated recently that "quantitative results in blood coagulation are not transferable between rats and humans" [5]. In wild animals, factors such as population density, stress, and hibernating also affect coagulation [8].

In addition to the reported species differences in blood coagulation, we add data to clot formation and the behaviour of fully formed clots, which comes after thrombin generation. The relationship with the preceding step lies in the influence of thrombin, fibrinogen, and calcium concentrations on clot architecture [16] and on the activation of factor XIII to stabilize the clot. Viscoelastic hemostatic assays obtain the kinetics of clot formation, and available data confirm the hypercoagulable status of rat blood (compared to human), expressed by a short clotting time, a high clot-forming rate, and a high clot stiffness [17,18]. The determining factor to rat blood clots' stiffness is certainly the high platelet count. Platelets stiffen clots because they incorporate firmly into the fibrin fiber meshwork through the covalent binding of their activated $\alpha_{IIb}\beta_3$ integrin to the distal end of the fibrinogen molecule and to more central sides on the $\alpha$-chain [19,20]. Platelets thereby create rigid building blocks within the meshwork, which not only make the clots stiff on macroscopic scale but also hinder compensatory local changes in network geometry when the fibers are strained. However, the platelet count is not the only cause for high clot stiffness in rats. In a previous test series, we reduced the platelet count by mixing a small volume of platelet-rich plasma in autologous platelet-depleted plasma, and we compared outcomes to samples from other species that had comparable platelet counts. Even in this condition, fibrin clots prepared from rat plasma were less compliant than those from humans, horses, and pigs [21]. The extraordinary stiffness of rat clots must therefore combine network effects (branchpoint density and dimension, branchpoint size and stiffness, distance (fiber length) between branchpoints) with fiber effects (fiber bundle distensibility). This suggests an adverse behavior of rat blood clots compared to human when stretched beyond their elastic limit. The slightly increased plasma concentration of factor XIII [7] in rats also stabilizes the fiber bundles, since whole blood forms clots with coarse bundles of varying thickness and cross-linking. The incorporated RBCs elevate also clot stiffness by adding their moduli, but they bind differently than platelets to the network [22] and block deformations [21,23,24]. Due to the importance of rats in experimental research and toxicology testing, we investigated differences of clot formation out of whole blood in some laboratory rat strains also considering the effect of sexual dimorphism. In a further test series, we analyzed the deformability of clots at large oscillation amplitudes (large amplitude oscillation; LAOS) by using a recent protocol for clot phenotyping by rheometry [21]. With the mechanical response that arises from slowly elevating the dynamic shear stress, it is possible to detect local structural arrangements within the sample. Tests are finalized when the clot is strained out so much that it breaks apart. Thrombelastometry, being the global clotting test used in clinical practice, does not provide this information. The aim was to determine whether wild-type laboratory rats are suitable animal models for clot behavior in the case of large dynamic deformations, such as those found in the vascular system.

## 2. Materials and Methods

### 2.1. Animals

A total number of 40 adult rats of 10–16 weeks old (each animal originating from a different litter) inbred and with wild-type genome were included in the study. We used 21 HL rats (for information about this hairless inbred strain see [25]) (8 females for test series one; 13 animals of both sex for test series two), 14 mutant Dahl/SS/JrHsdMcwiCrl rats (8 females for test series one, 3 males and 3 females for test series three), and 5 male

OFA rats (Crl:OFA(SD)) for test series three. For validation, test series three was conducted two years later with the F14 inbred generation of the originally purchased Dahl/SS rats. All animals were kept conventionally in polysulfone cages mounted by filter tops on a 12 h day/12 h light cycle in an animal house of the Medical University Vienna and received conventional diet (extruded: V1326-00; Ssniff, Roetgen, Germany) and tap water ad libitum. The health status was recorded on a regular basis, and animals were free of ectoparasites and worms but were positive for protozoa (*Spironucleus* sp.). They were also negative for viruses except *Murine Norovirus* but colonized by *Pasteurella pneumotropica*, *Helicobacter* sp., *Staphylococcus sciuri*, and *Corynebacterium bovis*. OFA rats originated from a SPF facility of the Medical University of Vienna. All rats were controlled by the veterinarian prior to the experiment and considered clinically healthy. They showed no signs of inflammation. The animals were kept in accordance with institutional policies and federal guidelines.

### 2.2. Blood Samples

Blood was drawn from rats in general anesthesia (100 mg/kg ketamine + 5 mg/kg xylazine i.p.) by inserting a 21 gauge needle into the cardiac ventricle under smooth suction by hand. The first millilitre of blood was used for hematology or was discarded, and the needle was mounted thereafter with a 5 mL syringe to collect the blood for the clotting tests. Blood from each individual was quickly put into a 3.2 mL tube containing 3.8% sodium citrate for anticoagulation (Greiner Bio-One, Kremsmünster, Austria) and gently tilted upside down to mix it with the anticoagulant. Blood began to be processed 20 min after withdrawal due to transportation from the animal house to the laboratory. If small inhomogeneities were visible after that time of transportation, the sample was discarded. Results were compared with human reference data [22] (in test series one) or were compared to eight healthy volunteers (in test series three) whose blood was drawn after giving informed consent (age: 55–62 years, 3 non-smokers, 5 smokers) using a Vacuette blood collection system containing 3.8% sodium citrate for anticoagulation.

### 2.3. Rheometry

A Physica MCR 301 rheometer (Anton Paar, Graz, Austria) was used. Temperature was Peltier controlled and set to 37 °C. A tempered hood mounted the measuring system, and a silicon oil filled evaporation blocker prevented sample drying. In the first and second test series, a profiled stainless-steel plate–plate system (PP25/P3, 25 mm diameter, gap width 1 mm) was used. The system was filled with 580 µL citrated whole blood after re-calcification with 38.5 µL of 0.2 M $CaCl_2$ solution (TEG® Hemostasis System, Haemonetics, Boston, MA 02110, USA). In the third test series, a sand-blasted stainless-steel cone–plate measuring system (50 mm diameter, 1° cone angle, 0.1 mm tip truncation) was used to obtain the non-linear behavior of the fibrin gels. The cone–plate system had to be used to ensure a stabile sinusoidal shear stress input to the sample at large amplitudes. The system was filled with 580 µL blood after re-calcification with 40 µL of 0.2 M $CaCl_2$ solution. Time sweeps were conducted at constant frequency (1 Hz) and low deformation amplitude (first series: 0.001%; other series: 0.01%) to ensure only minimal mechanical interference with the ongoing clotting process according to [26,27]. The developing shear moduli (storage modulus G′, loss modulus G″) were calculated from the stress ($\tau(\omega)$)/strain ($\gamma(\omega)$) relationship by using the shift of the phase angle ($\delta$): G′ equals $\tau(\omega)/(\gamma(\omega))$ * cos($\delta$); G″ equals $\tau(\omega)/(\gamma(\omega))$ * sin($\delta$). Completed clotting was defined when G′ reached a plateau value (G′$_{plateau}$). For the large amplitude oscillation (LAOS) stress protocol (LAOStress) in test series three, we set several logarithmic shear stress increments from 1–5,000 Pa at constant angular frequency ($\omega$, 1 rad s$^{-1}$) and measured the strain response of the clot out of the oscillation cycle for each stress level. The cyclic stress–strain dependency is displayed by Bowditch–Lissajous plots, from which we gained the clot compliances (J′-moduli) that gave rise to the three shear stress thresholds that we obtain from cyclic loading: (1) onset of irreversibility ($\tau_D$), (2) onset of microscopic stiffening; the clot deforms non-affine ($\tau_L$),

and (3) onset of macroscopic stiffening, which is the threshold at which the clot starts with affine deformation ($\tau_M$), as described recently [21,28].

### 2.4. Experimental Protocols

In the first test series one portion of the whole blood sample was allowed complete clotting. The kinetic test was stopped at +45 min. Thereafter, a frequency sweep test was started to investigate the linear behaviour of the clot while the clot remained in the test geometry. Thereafter, the plate with the clot adhering on it was removed from the rheometer, rinsed in phosphate buffer (without $Ca^{2+}$ and $Mg^{2+}$, pH 7.4) and immersed in 3.5% formaldehyde solution for 30 min. Meanwhile, a new sample was filled into the rheometer gap by using an additional plate, and clot formation was stopped at +15 min. The plate with the sample adhering on it was rinsed as well and immersed 30 min in formaldehyde for in situ fixation. Both clots were carefully removed with a surgical knife from the plate, rinsed again in phosphate buffer, and afterwards dehydrated in stepwise manner in ethanol–water solutions (30, 50, 70, 90, 98% ethanol). Samples were incubated in each ethanol-water solution for 20 min and finally stored in 98% ethanol until assayed by SEM. In parallel, thrombelastometry was performed from each blood sample in technical duplicates.

In the second test series, the effect of sex was tested. Eight male and thirteen female HL rats of 10–16 weeks of age were used. Stiffness of clots ($G'_{plateau}$) was associated with platelet count and hematocrit of a paired EDTA-sample.

In the third test series, whole blood samples were allowed completed clotting like in the other test series and subjected thereafter to the LAOStress protocol while remaining in situ [21].

### 2.5. Other Laboratory Data

The hematological profile was performed with ADVIA 2120i (Siemens, Munich, Germany). Plasma fibrinogen concentration (FIB) was measured by the method of Clauss. Thrombelastometry was conducted in technical duplicate and triplicate using the ROTEM® delta (TEM International, Munich, Germany) in NATEM mode at 37 °C for 1 h according to the manufacturer´s specifications. Samples were re-calcified with 0.2 M $CaCl_2$-solution (TEG® Hemostasis System, Haemonetics, Boston, MA, USA) using the built-in auto-pipette.

### 2.6. Scanning Electron Microscopy (SEM)

Dehydrated clots were dried in a vacuum oven for 30 min at 40 °C and gold coated (Scancoat Six; Edwards, UK). SEM was operated under high vacuum conditions and with variable voltages between 5 and 20 kV using a Quanta 250 FEG SEM (Thermo Fisher Scientific, Waltham, MA, USA). A secondary electron detector was used to generate the micrographs.

### 2.7. Statistic

The expected difference between human and rat values and ethical considerations allowed us to limit the sample size to a minimum. A power analysis for comparing mean values of rats versus humans in regard to clot stiffness ($G'_{plateau}$) and shear stress thresholds ($\tau_D$, $\tau_L$, $\tau_M$) (own pilot experiments) resulted in a number of 5 individuals per group (power 0.8; two-tailed). For detecting a difference of $G'_{plateau}$ between rat strains, a number of 8 animals per strain was calculated. Due to this low sample size, data are presented as medians with the quartiles in brackets. The rheological parameters (G', $\gamma(\omega)$, $J'_M$, $J'_L$, R) were exported from the Rheocompass software (version 1.19, Anton Paar, Graz, Austria). Other laboratory data were added manually to the data file. Data were processed in GraphPad Prism (version 8.1; GraphPad, La Jolla, CA, USA) on MacOS Sierra. Clot stiffness was identified by $G'_{plateau}$ in rheometry and by MCF (maximum clot stiffness) in thrombelastometry. The difference of clot stiffness between rat strains was analyzed by the two-tailed Mann–Whitney test. The difference to human clots is only shown graphically.

$G'_{plateau}$ values were correlated to MCF, platelet count (PLT) and hematocrit (HCT) values by Spearman correlation. For the nonlinear behavior of clots, we manually obtained three shear stress thresholds from the compliance curves. These thresholds were obtained by extrapolating the J´-moduli´s divergence ($\tau_D$) and maxima ($\tau_L$, $\tau_M$) to the shear stress axis (*x*-axis). We also obtained the maximum shear stress, the maximum shear strain, and the maximum G´ before clots broke apart or were detached from the plates. Rat clots were compared to human clots by using the two-tailed Mann–Whitney test. To show the sustainability of clots, Kaplan–Meier curves were plotted by using the maximum shear stress that the individual clot could sustain to estimate the survival function of clots from the species (displayed as percentage of clot survival).

## 3. Results

### 3.1. First Test Series

Clotting differed between the HL and Dahl/SS rats, as it started later but gained higher $G'_{plateau}$ values in Dahl/SS rats (HL: 609 (582/704) Pa vs. Dahl/SS: 826 (729/1011) Pa; $p < 0.001$). When the clotting started (+15 min), fibrin fibers were first straight and left pores open, but later (+45 min), the fibers became highly contorted and the network dense. A great difference exists between the kinetics of clot formation between rats and humans (Figure 1). Figure 2 shows the difference of rat and human fiber architecture. Frequency sweep tests (Figure 3) showed almost no frequency dependence of fully formed rat clots. Clots remained in their equilibrium over the entire frequency range and oscillated as a whole in the gap. The same occurs generally with human clots (personal communication). A G″-minimum was visible between 10 and 20 rad s$^{-1}$, indicating highest clot elasticity around a 100 beats per minute heart rate. The elasticity of the initial network with straight fibers was more frequency-dependent and therefore not as stabilized (Figure 3a). Thrombelastometry confirms the higher clot stiffness of Dahl/SS rats, but the difference between the two strains was less pronounced (Figure 4). MCF and MCF-t were higher in Dahl/SS rats (MCF: 73 (70/75) mm; MCF-t: 34 (33/35) min) than in HL rats (67 (64/69) mm; MCF-t: 31 (25/32.5) min; $p < 0.05$), showing that the stiffer Dahl/SS clots also needed more time to be fully formed. The alpha angles were indistinguishable, and the clotting time showed a trend to be higher in Dahl/SS rats, which confirms rheometry. There was a correlation between clot stiffness obtained by rheometry ($G'_{plateau}$) and thrombelastometry (MCF) only in the Dahl/SS strain (r = 0.86; $p = 0.01$). We conclude that clot stiffness is more than doubled in rats compared to humans and also differs between rat strains. Thrombelastometry underestimated the differences in clot stiffness among rats.

### 3.2. Second Test Series

$G'_{plateau}$ values were higher in male HL rats (1025 (892/1115) Pa) than in female HL rats (685 (610/768) Pa; $p < 0.001$, Figure 5), although platelet count was not different between the sex groups (males: 1016 (964/1059) G/L; females: 950 (904/1015) G/L). Platelet count therefore did not correlate with clot stiffness ($G'_{plateau}$), neither in the sex groups' (a trend in the male group was present; Spearman r = 0.64; $p = 0.69$) nor in the combined values from both sexes (Spearman r = 0.23). Likewise, HCT was indifferent in the sex groups (males: 43 (42/44.8) %; females: 42.1 (41.4/44) %), and there was no correlation between $G'_{plateau}$ and HCT. Other factors than platelet count alone make clots of male rats stiffer.

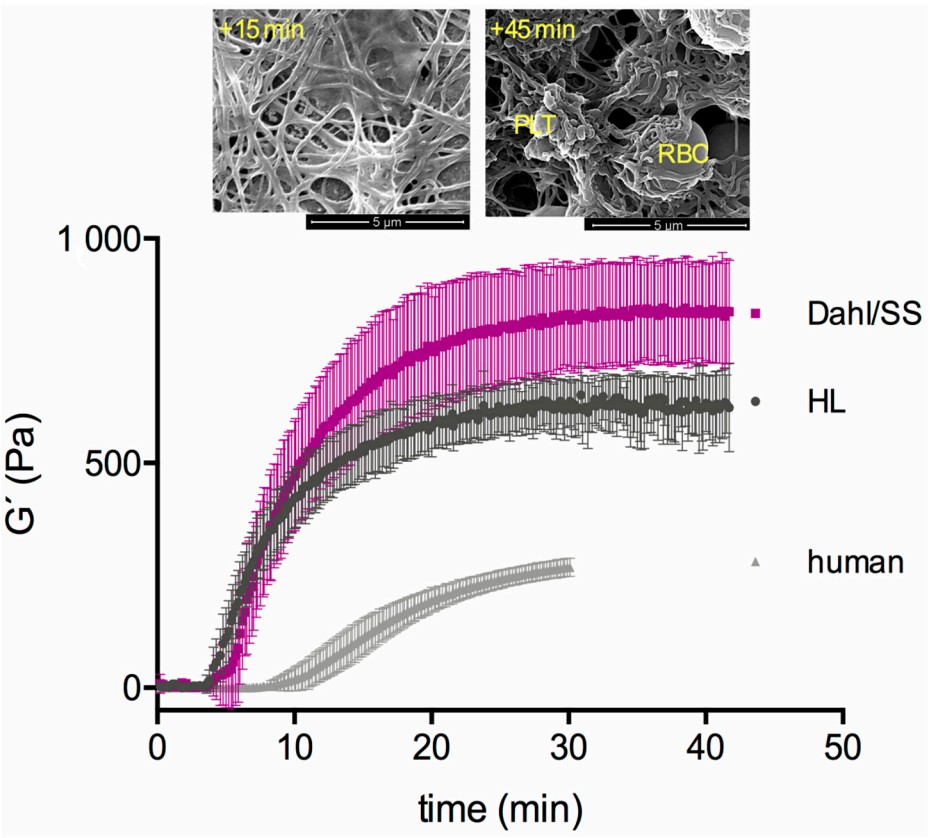

**Figure 1.** Clot formation in female HL (grey) and female Dahl/SS (purple) rats (clot formation in human males (in light grey color) are included from our previous study [22] as reference to highlight the pronounced clot stiffness of rats. The final clot stiffness (= G′-plateau value) is higher in the Dahl/SS strain than in the HL strain. SEM images highlight the morphology of rat clots at +15 min and at the end of the kinetic test. Whereas fibrin fibers in the network are initially straight, they become contorted after 45 min.

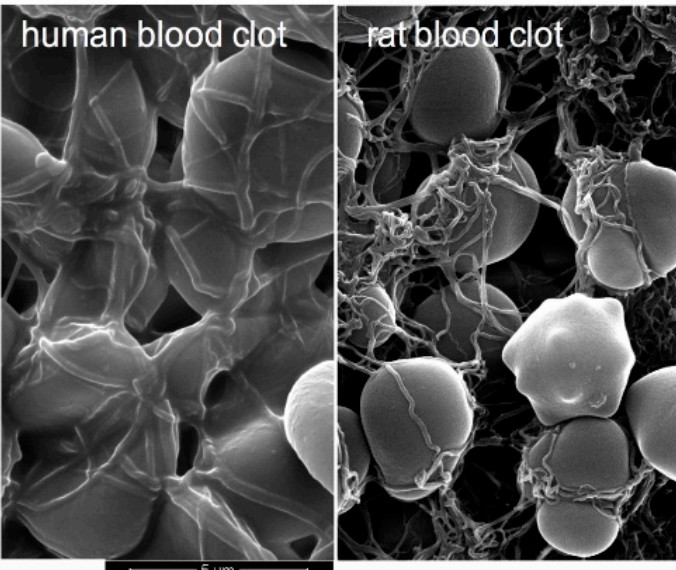

**Figure 2.** SEM of fully formed human and rat blood clots at 15,000× magnification. Rat fibrin fibers are thinner and more contorted, even if they bind to RBCs, and the fiber meshwork is denser. Note that the human RBCs are species-specifically larger.

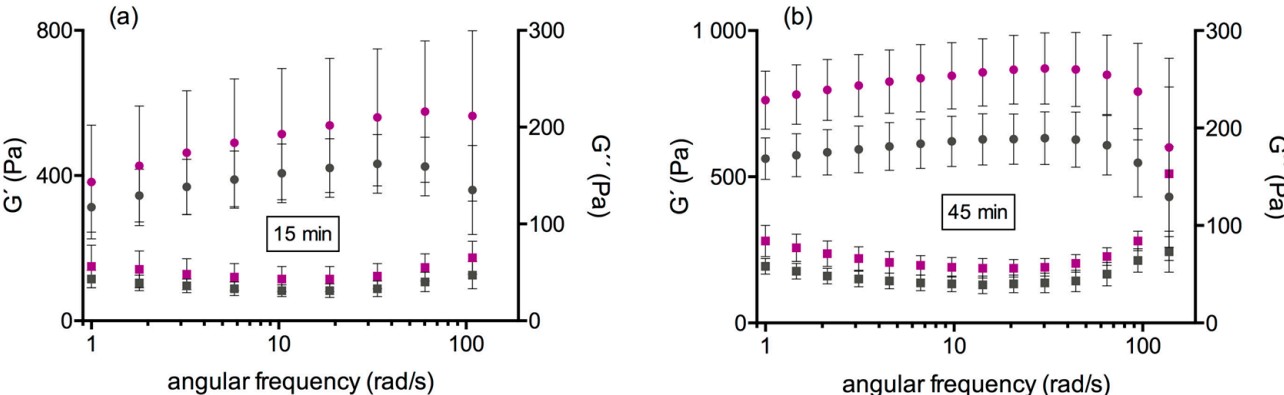

**Figure 3.** Frequency spectrum of storage (G') and loss modulus (G") at low deformation ($\gamma(\omega) = 0.01\%$): (**a**) at +15 minutes following start of clotting, (**b**) at +45 minutes following start of clotting. Although the shear moduli have higher values in the Dahl/SS strain (purple) compared with the HL strain (grey), the linear behavior of the clots is similar. In all cases, G" displays a minimum between 10 and 20 rad/s.

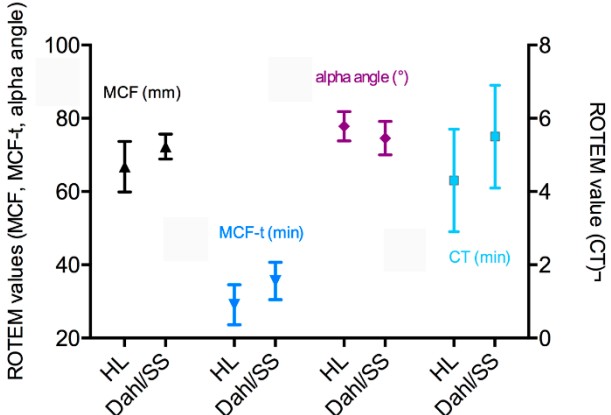

**Figure 4.** Thrombelastometry of clots from HL and Dahl/SS rats.

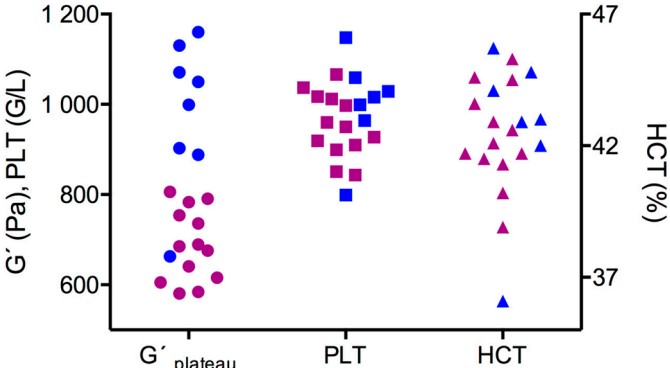

**Figure 5.** Stiffness of fully formed clots (G'$_{plateau}$) and hematological values in the sex groups (blue: males; purple: females). Clot stiffness was higher in male rats, but platelet count (PLT) and hematocrit (HCT) values of males and females overlapped.

### 3.3. Third Test Series

Laboratory values were for Dahl/SS rats: PLT: 304 (268/362) G/L; FIB: 254 (251/307) mg/dL. For OFA rats, they were: PLT: 758 (693/849) G/L; FIB: 299 (279/310) mg/dL, and for human volunteers: PLT: 112 (95/149) G/L; FIB: 376 (322/431) mg/dL. Figure 6 shows the frustrated attempts of rat clots to shear-stiffen. Instead, the clots displayed a prolonged

softening up to about 200 Pa shear stress, in which they lost a third of their strength, in contrast to the human clots that started to stiffen beyond 100 Pa shear stress with very little preceding softening. Clot strength at maximum stretch-out therefore narrowed in the species but remained higher in OFA rats (Dahl/SS: 429 (396/490) Pa (n.s. compared to human); OFA: 538 (515/576) Pa ($p < 0.01$ compared to human); human 396 (345/437) Pa). All shear stress thresholds moved to higher values in clots from the rat strains ($p < 0.005$), and $\tau_M$ was even absent (only a discrete shoulder was observed in OFA). This indicates that the behavioral switch from non-affine to affine deformation was not achieved. The onset of irreversibility due to cyclic loading ($\tau_D$) was also delayed. Specifically, values in Dahl/SS rats were: $\tau_D$: 78 (57/122) Pa, $\tau_L$: 427 (382/473) Pa, $\tau_M$: n.a. In OFA rats, the values were: $\tau_D$: 133 (197/133), $\tau_L$: 586 (474/587), $\tau_M$: n.a.; and in humans: $\tau_D$: 22 (20/35) Pa, $\tau_L$: 181 (90/201) Pa, $\tau_M$: 426 (293/472) Pa. Clots from Dahl/SS rats could sustain higher shear stresses before they broke apart (895 (893/1177) Pa) compared to humans (722 (532/853) Pa), also reaching higher maximal shear strains (218 (205/231) %, compared to human 182 (165/192) %; both $p < 0.05$). Clots from OFA rats likewise sustained higher shear stresses (1112 (812/1376) Pa; $p < 0.05$) but displayed similar maximal shear strains (204 (176/229) %) like human clots. Median survival was at 895 Pa for Dahl/SS rats, 1112 Pa for OFA rats, and 723 Pa at human rats ($p < 0.05$; Figure 7). We conclude that rat clots are more than twice as stiff as human clots in quasi-static condition but lose much of their stiffness when they are strained out (rat clots yield). Despite the preceding yielding, rat blood clots broke apart macroscopically at 25% higher loads than human blood clots, having also 15% higher shear strains. Inbreeding Dahl/SS rats in our animal house affected clot stiffness. $G'_{plateau}$ values decreased by 23% (from 826 (729/1011) Pa in test series one to 632 (624/671) Pa in test series three; $p < 0.001$), reflecting a phenotypical deviation.

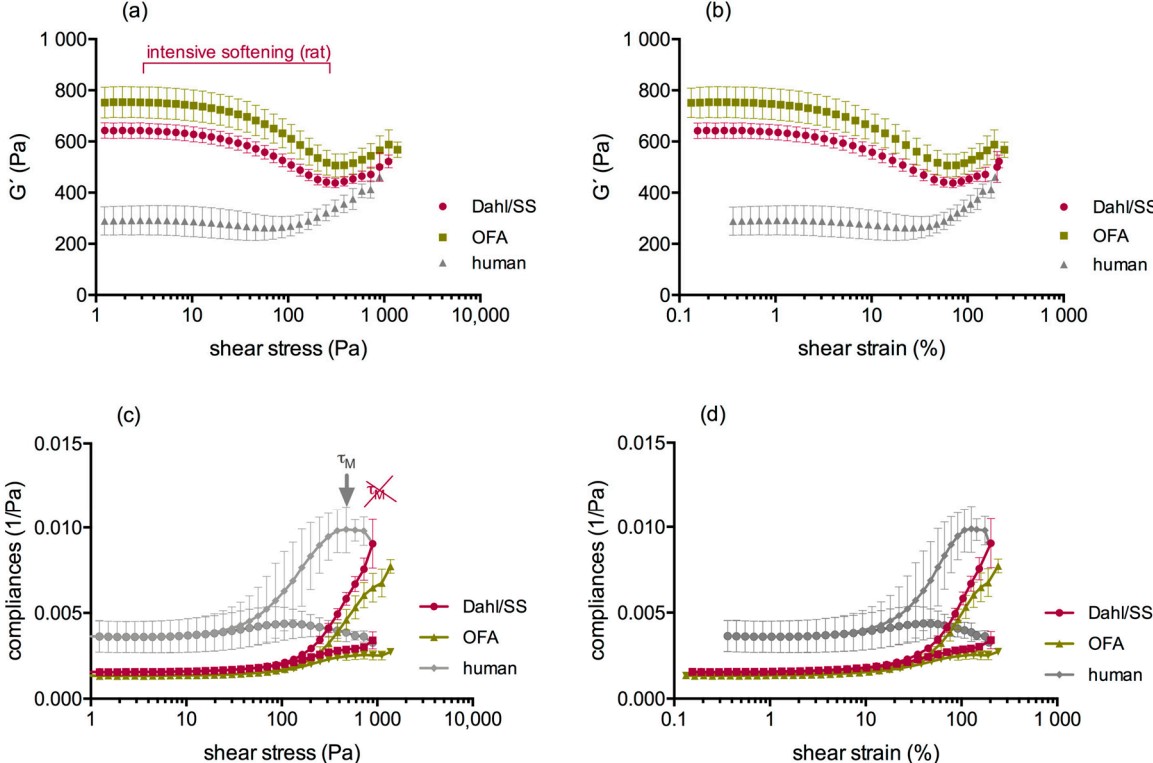

**Figure 6.** LAOStress tests of whole blood clots from rat strains (purple: Dahl/SS, green: OFA) compared to human (grey). Rat clots showed a phase of intensive softening before they started to shear stiffen (**a**,**b**). However, G′ did not reach initial values again. In contrast, human clots started to shear-stiffen at lower shear stress, and G′ raised progressively. In contrast to human clots, rat clots could not align all fibers in the direction of the shear (**c**,**d**) $\tau_M$ is lacking in rats.

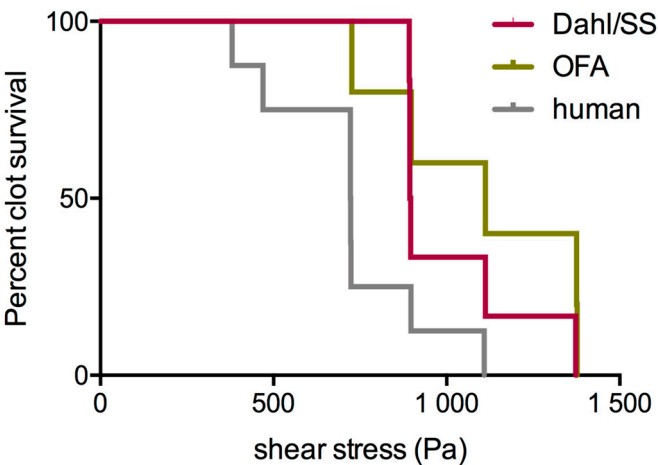

**Figure 7.** Rat clots withstood ("survived") high shear stresses better than human clots.

## 4. Discussion

All test series showed that rat whole blood clots are stiffer than human whole blood clots at quasi-static condition, as they exhibited two- to threefold higher G'-values. Clotting also started in half the time, and clots formed much quicker in rats than in humans, which impressively demonstrates the hypercoagulability of rat blood [4]. Already 15 min following start of the coagulation cascade, a Mikado-like network of straight fibers was formed, which remodeled thereafter into a dense meshwork of contorted fibers (compare with Figure 1 and [17]). This highly elastic fiber geometry (mean G' between 600 and 800 Pa) initially resisted being deformed, but the network expanded well before it broke apart (compare with Figure 6b,d), which is not plausible from its density (Figures 1 and 2). There is a delicate difference in the clot behavior in LAOS between rat and human clots, which must be considered for the explanation of this high strain. While human clots typically stiffen as shear stress increases, rat clots do just the opposite (Figure 6a). Rat clots became successively compliant with increasing shear deformation (J'-moduli only increased), reducing their strength by one-third. Why rat clots softened that much can only be speculated. Either the thinner or shorter fibers of the meshwork broke, leaving bundles of greater length and thickness. The microfractured clot then begins to stiffen at higher shear stress as the remaining bundles are now forced to deform. Alternatively, the contorted fibers straightened (supported by weak coupling at the branchpoints). A weakening would then result from fiber bending and the disentanglement of the bundles [29]. In both cases, a certain fiber length is "gained" irrespective of how this gain is achieved. This increase in length explains the achieved clot deformation. Supportive for the first option is the absence of $\tau_M$ (compare with Figure 6c). This means that the fibers could not fully align [30]. If the meshwork would be more mobile, it could shift the strain from regions of high deformation to regions of low deformation and thereby arrange all fibers in a way to distribute the load equally among them. In contrast, when the fibers struggle to stretch, the most stressed fibers fail first (those with existing pre-stress, e.g., bound to platelet filopods), causing the stress to propagate in the remaining load-bearing fibers until they also break. This reduces clot strength, which we observed [31]. The reported higher factor XIII plasma concentration of rats [7] certainly contributes to bundle stiffness through the covalent crosslinks between the assembled fibrin chains.

Platelets are decisive for clot stiffness because they are firmly bound to fibrin fibers [32–34]. Incorporated platelets (platelet aggregates) are themselves stiff and elevate the modulus of a clot by quantitative means. They are not passive, but instead pull on fibers through their ability to contract, thereby giving them a pre-stress [35], which enhances stiffness by qualitative means. The high G'-moduli of rat clots at small deformation are in large part the result of the high physiological platelet count of this species. The distribution of large numbers of platelets within the clot retards the elongation and

alignment of the fibers due to the prestress they exert. The high platelet concentration is therefore in large part also responsible for the clot behavior at large deformations. Compression experiments showed that higher moduli are gained from platelet-rich clots and lower moduli from RBC-rich clots [36], which we have confirmed by shear experiments [22]. One should not neglect that RBCs also play a role for clot stiffness [37]. If they are added to a plasma sample, they elevate the shear elastic modulus [24]. However, when the clot is deformed beyond its elastic limit, they block fiber lengths by binding to fibrin, and they make fiber alignment and platelet contraction more difficult [38]. In this study the RBCs alone cannot be held responsible for blocking the shear stiffening response of the fibers since humans had a comparable hematocrit and the human fibers were able to fully align beyond a mean shear stress of ≈400 Pa. We showed recently that even if rat clots contain no RBCs and only few platelets (10 G/L), they can hardly stiffen with shear [21]. A similar behavior can be found in human clots only at platelet counts that are ten times higher. This points to a relative inflexibility of fibrin fiber bundles and again supports the first option for softening. Available information on the composition of rat fibrinogen is limited, but differences can be mostly expected in the α-chain since this is the least preserved chain [39]. In fact, rat fibrinogen has a shorter α-chain (and a shorter β-chain) compared to the human analogue, and it contains less tandem repeats; therefore, the tethers in the αC-polymer must be shorter [40]. Since fiber distensibility strongly varies with the length of its tandem repeats [41], rats might actually have less flexible fibrin fibers, which again supports the first option for clot softening.

Clot elasticity also differed between rat strains, with OFA and Dahl/SS rats showing higher G′-values than HL rats, but the general behavior at LAOS was similar, suggesting it is a general feature of rat blood clots. The comparison of rheometry with the clinically used thrombelastometry showed good correlation for clot stiffness values (G′$_{plateau}$ vs. MCF), but with thrombelastometry, the differences between rat strains could not be represented as well as with rheometry. The explanation for this finding may lie in the different oscillation amplitudes of the two systems. The shear condition during clot formation alters clot configuration and, as such, also the viscoelastic parameters obtained [42,43]. The ROTEM equipment that we used works at relatively high strain amplitudes that generate a significant deformation in the clot in the gap [19]. By rheometry, much lower shear strains can be applied (in our tests: ≤0.01% = 100 nm) so that the clots developed at minimal mechanical disturbance. Higher strains mask the true kinetics of clot formation. Clot stiffness also differed with sex. Male HL rats had stiffer clots even though the platelet count and fibrinogen concentration were indistinguishable between males and females. Rat platelets might be very susceptible to androgenic stimulus [44]. This should be considered for animal selection.

For the species of rat, it must be questioned whether the hypercoagulability and the clot behavior shown here are an advantage or not. Certainly, the short clotting time and the high clot stiffness protect rats from severe bleeding. This benefit is offset by microfractures in the material as soon as clots must stretch out. Such strain amplitudes occur especially in the arterial tree due to the longitudinal pressure propagation and the pulse wave. If fractures are locally concentrated, sheets of the clot might break off and be washed away. But small and homogenously distributed fractures can open pores in the meshwork, which allow the penetration of plasminogen activators. In general, whether this specific clot phenotype is an advantage or a disadvantage for the species depends on the in vivo composition of rat clots, which can vary significantly with regard to the site of origin, at least in human [45]. Nevertheless, the clot behavior differs fundamentally from that of humans and must be taken into account if the mechanical behavior of clots is the determining factor in studies.

Limitations of the study: By the use of native blood and a global clotting test, numerous factors involved in thrombin generation and platelet activation were not analyzed or standardized. In addition, the onset of fibrinolysis during the LAOStress test cannot be excluded. Previous tests with native whole blood rat clots, however, showed that the G′

value remained constant for more than 30 min once it had gained its plateau value when the shear conditions (low amplitude, the clot remained in its linear state) were maintained. This may be due to the resistant murine fibrinolytic system to its activation [11]. In this regard, it must be considered that the percent fibrinolysis that is displayed in ROTEM® tests after reaching the MCF value does not solely reflect fibrinolysis but also includes gradual clot shrinkage with syneresis and nonlinear (plastic) deformation due to the large oscillation amplitude applied. This explains the high standard deviations in this parameter [15]. The genetic background of the rat strains (especially the HL strain) was not identified prior to the experiment. The findings are also limited to the age of animals (10–16 weeks) and the strains used.

## 5. Conclusions

In clots generated in vitro from native blood of rats by addition of stoichiometric calcium to citrated samples, the fiber response is massively blocked by (1) numerous platelets dispersed throughout the clot creating a pre-stress for the fibers, (2) the presence of RBCs blocking fiber alignment, and (3) the relative rigidity of the fibers based on fibrinogen chemistry. These factors combine to soften the clot to two-thirds of its initial modulus when deformed similar to conditions in the arterial bed. There remains only a frustrated attempt to shear-stiffen before the clots break apart. Cause for the observed yielding will be microfractures in the fiber meshwork, which at same time open pores for dissolved substances. This behavior contrasts with the behavior of human blood clots and renders rats unsuitable as experimental animals when the mechanical behavior of clots is the dominant factor in studies. Because we used three wild-type strains (including one mutant strain), this result applies to healthy wild-type laboratory rats in their entirety, although strain-specific and sex-specific differences are present.

**Author Contributions:** Conceptualization and methodology, U.W.; investigation, V.G., L.P., U.W.; resources, U.W.; writing, U.W., V.G.; visualization, U.W.; project administration, U.W. All authors have read and agreed to the published version of the manuscript.

**Funding:** This research received no external funding.

**Institutional Review Board Statement:** The animal study protocol was approved by the Institutional Ethics Committee of the Medical University Vienna and by the national authority (BMWF-66.009/0284-V/3b/2019, BMWF-66.009/0085-V/3b/2018). Blood collection from humans was conducted in accordance with the Declaration of Helsinki and was approved by the Ethics Committee of the Vienna Hospitals of the Vinzenz Group (EK09/2019); part of control group data that were available at time of this study were used. Informed consent was obtained from all subjects involved in the study.

**Informed Consent Statement:** Informed consent was obtained from all subjects involved in the study.

**Data Availability Statement:** Data can be accessed from the authors on request.

**Acknowledgments:** We thank Attila Kiss for supporting us with residual blood samples from OFA rats.

**Conflicts of Interest:** The authors declare no conflict of interest.

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
