# Peer review of "Laboratory Rat Thrombi Lose One-Third of Their Stiffness When Exposed to Large Oscillating Shear Stress Amplitudes: Contrasting Behavior to Human Clots"

_2673-8937, doi:10.3390/ijtm2030026_

Round 1

Reviewer 1 Report

The work of Windberger et al. is devoted to understanding of mechanic aspects of rat thrombi and their comparison to human thrombi. The authors study different strains of rats and unveil a plethora of specific features of rat thrombus mechanics that should be at least taken into account before using rats for hemostasis/thrombosis studies. Although, the manuscript of Windberger et al. is of significant interest to the field, some  issues should be addressed before the work can be accepted:

Major points

1.      Prior to studying of the mechanical properties of the blood thrombi of rats and comparison of them to the human ones, more conventional parameters of the rat blood coagulation should be assessed (e.g. fibrinogen concentration, typical plasma coagulation times, fibrinolysis parameters). These data can provide deeper insight into the meaning of differences between human and rat blood

2.      The authors hypothesize that major differences might be explained by the alterations in the fibrin structure in rats. However, blood clot stability is also determined by platelet count and function. Furthemore, in human platelet contraction is an important mechanism of blood thrombus stabilization. Are there any differences between human and rat platelets?

3.      It is of interest to see whether depletion of the blood platelets from the system (e.g. collection of the PRP from whole blood upon centrifugation, separation of platelets and platelet poor plasma and addition of platelet poor plasma to RBC) affects thrombus stability in rats and humans.

4.      Observed significant differences between different sex groups are poorly addressed. What are the potential explanations of this phenomenon?

Minor points.

1. The statement that the clot stiffness is dependent on platelets is not obvious and is based only on the previous works of the same authors. This statement should be explained in more details.

2. Methods utilized for clot formation and analysis should be listed in the Abstract.

3. Although I don't feel qualified to judge about the English language and style, I would suggest extensive english editing, as the meaning of some of the sentences is unclear. For example, in the Abstract: "platelets also import numerous stiff branchpoints into a clot´s fibrous system", 

Author Response

Major points

  1. Prior to studying of the mechanical properties of the blood thrombi of rats and comparison of them to the human ones, more conventional parameters of the rat blood coagulation should be assessed (e.g. fibrinogen concentration, typical plasma coagulation times, fibrinolysis parameters). These data can provide deeper insight into the meaning of differences between human and rat blood

Answer: we rearranged the introduction section that now contains this information.

  1. The authors hypothesize that major differences might be explained by the alterations in the fibrin structure in rats. However, blood clot stability is also determined by platelet count and function. Furthemore, in human platelet contraction is an important mechanism of blood thrombus stabilization. Are there any differences between human and rat platelets?

Answer: the importance of platelet count is already described in the initial submission in the second paragraph of the discussion section. Platelet contraction certainly contributes to the prestress of fibers (see our description on page 7) and species-specific differences are obvious: rat platelets have different amount of contractile proteins and myosin ATPase (https://doi.org/10.1016/0049-3848(79)90117-8). A sentence was included to the introduction section of the revised version concerning these differences between rat and human. We also added the words “platelet contraction” to the discussion section where we describe the fiber prestress.

  1. It is of interest to see whether depletion of the blood platelets from the system (e.g. collection of the PRP from whole blood upon centrifugation, separation of platelets and platelet poor plasma and addition of platelet poor plasma to RBC) affects thrombus stability in rats and humans.

Answer: we have already depleted platelets from the system and tested human PDP samples and one rat PDP sample in a pilot study (see the figures below). We found a significant difference between human and rat clot behavior pointing to the importance of the fibrinogen network architecture to clot strength (human and the OFA rat had similar fibrinogen concentrations and tests were started in Natem mode). Therefore, not only the platelets and their contraction are responsible for the differences between human and rat clot behaviors.

What can also be postulated from these figure is the fundamentally different behavior of fibrin clots versus whole blood clots (in humans AND rats). When the platelets and the RBCs are removed the fiber system can indeed shear-stiffen, but the compliance of the rat system is still much lower, and the meshwork also needs higher shear stresses to be re-arranged. These data support our hypothesis (in the discussion section) of reduced fiber distensibility in rat fibrin networks. We described the effect of RBCs on clot behavior already (10.3390/molecules25173890). Here, we now added sentences concerning the relevance of RBCs to clot behavior in the revised version in the introduction and discussion sections.

  1. Observed significant differences between different sex groups are poorly addressed. What are the potential explanations of this phenomenon?

Answer: an explanation has been included in the discussion section (androgenic stimulus)

Minor points.

  1. The statement that the clot stiffness is dependent on platelets is not obvious and is based only on the previous works of the same authors. This statement should be explained in more details.

Answer: please see one previous paper from 1978 (PMID: 694229): Res Commun Chem Pathol Pharmacol. It has long been known that platelets are most responsible for clot strength. Here also a recent one: DOI: 10.1161/STROKEAHA.121.035105. Platelets do this by “freezing” the system. Platelet aggregates increase the stiffness of the clot because they act as pre-stressed nodes. They import and distribute local stiffnesses into the fiber system. We explain this fact in the discussion section now more extensively.

  1. Methods utilized for clot formation and analysis should be listed in the Abstract.

Answer: has been added.

  1. Although I don't feel qualified to judge about the English language and style, I would suggest extensive english editing, as the meaning of some of the sentences is unclear. For example, in the Abstract: "platelets also import numerous stiff branchpoints into a clot´s fibrous system", 

Answer: we have changed this sentence accordingly: “Platelets also import numerous stiff junction points into the fibrous system of a clot“.We have asked a native speaker to help us with formulations, and the suggestions were included.

Reviewer 2 Report

The article entitled “Laboratory rat thrombi loose one third of their stiffness when exposed to large oscillating shear sress amplitudes: contrasting behavior to human clots” focuses on an important topic of research as the rats are used in many  experimental coagulation studies. The manuscript is well-written both in terms of laboratory methods and in terms of results. There is little research in this area. Therefore, I think that this manuscript is suitable for publication in its current version.

Author Response

Dear Reviewer,

thank you for your time that you spent on evaluating our manuscript, and thank you very much for your kind words.

Yours sincerely,

Ursula Windberger

Reviewer 3 Report

The study is clearly set out and supported by relevant methodological approaches. The data is well interrogated and interpreted, and the findings are clear and definitive. Principally, this is an observational study and highlights the fact that animal models do not provide, in many cases, clinically relevant data, endpoints and findings. I would have liked to have seen more mechanistic data or conclusions to the observations- e.g. genetic background of the murine strains. Have the authors looked at and further compared their results to other strains or indeed species. Was age taken into consideration? Do the authors have data on circulating Fibrinogen or vWF levels between strains? While not within the remit of this study, or scope, in future work it would be interesting to look at the ECM content/matrisome (quantitative and qualitative) in various clots and different timepoints. 

As it stands, it is fine for publication once more support is given in the discussion as to the observation and differences recorded.

Minor proof reading and editing would be good also.

Author Response

Answer: we thank the reviewer for her/his kind words! The introduction section has been revised to include information about differences in blood coagulation between rat and human (e.g., circulating coagulation factors, thrombin binding to platelets, etc.). These differences are now also used in the discussion section to explain some findings. They match nicely to what we have observed. A limitation paragraph is included in the discussion of the revised version and items like age and genetic background are described. Differences in platelet count and fibrinogen in the strains used and in human samples were already provided in the first submission (third test series). In the second test series platelet count and hematocrit was already provided.

We have tested other species as well (see the figure below and compare with figure 6c in our manuscript). Rat remains outstanding. The difference is not only to human clots but to clots of other mammals, too. Only cow clots come close to rat clots. Cow fibrinogen chemistry is known by crystallography. Fibers are relatively stiff, which supports our assumption on page 7. But these data are too few to show them in the manuscript (1 camel, 1 chicken, 5 cows, 3 horses).

Thank you for your suggestion to consider matrix effects. Environmental factors play a fundamental role in fiber assembly, and it is important to know what is in the “pores” of the meshwork.

Minor proof reading and editing would be good also.

Answer: we have asked a native English speaker to edit the text and have included her suggestions.

Reviewer 4 Report

With great interest I read the work of Windberger et al on the contrasting behavior of clots compared to human clots.

Authors should be congratulated on great and structured research ideas, identified research gap and work done. I would recommend the following to improve the quality of work:

Abstract: Please introduce the shortcut „LAOS“

Introduction: The last sentence is a summarization of your main findings. This does not belong to the introduction, but to the discussion. A clear statement of study aims is missing. Please revise.

Results:

Please avoid the conclusion in the results section. The results section is only to report on results not to discuss them. (Section 2.1 last 2 sentences).

Figure 2: The reader would more benefit from a double or three times sized picture presented.

Material and methods: In general nice description of used materials and applied methods.

4.1 Animals “10-16 weeks of age”, please reformulate into 10-16 weeks old or similar.

Limitations of the study as part of the discussion are missing. Please provide the limitations of the study and the methods used.

References and discussion of other studies comparing coagulation of rat and human blood or other animals are missing. Please add this to your discussion.

An organized conclusion is missing. Please provide a conclusion.

Author Response

Abstract: Please introduce the shortcut „LAOS“

Answer: has been added in the abstract and in the introduction section

Introduction: The last sentence is a summarization of your main findings. This does not belong to the introduction, but to the discussion. A clear statement of study aims is missing. Please revise.

Answer: the end of the introduction section was modified

Results:

Please avoid the conclusion in the results section. The results section is only to report on results not to discuss them. (Section 2.1 last 2 sentences).

Answer: we agree that in the results section only results should be described. However, we think also that it becomes easier for the reader to understand the results if we sum them up at the end with a few words. Therefore we like to keep these 2 sentences in the text.

Figure 2: The reader would more benefit from a double or three times sized picture presented.

Answer: I suggest, it concerns to figure 2? I inserted it in this scale due to space. I hope that all figures can be enlarged during processing. I shall submit all figures in original (bigger) version.

Material and methods: In general nice description of used materials and applied methods.

Answer: thank you!

4.1 Animals “10-16 weeks of age”, please reformulate into 10-16 weeks old or similar.

Answer: has been changed

Limitations of the study as part of the discussion are missing. Please provide the limitations of the study and the methods used.

Answer: has been added (please see page 8 of the revised version)

References and discussion of other studies comparing coagulation of rat and human blood or other animals are missing. Please add this to your discussion.

Answer: thank you for this comment. In regard to the comments of reviewer 1 we included this information to the introduction section of the revised version.

An organized conclusion is missing. Please provide a conclusion.

Answer: has been added